# OpenReview forum: "Pandora's Box: Towards Building Universal Attackers against Real-World Large Vision-Language Models"
_NeurIPS.cc/2024/Conference — NeurIPS 2024 poster_

### Official Review · Reviewer_i8wm · 2024-06-27

**Soundness:** 3
**Presentation:** 3
**Contribution:** 3
**Rating:** 6
**Confidence:** 3

**Summary:**

This paper studies the universal adversarial attacks on large vision language models (LVLMs). It focuses on the black-box setting where only the model's response is available to the adversary. This paper proposed a query-based targeted attack that leverages a proxy model to obtain the similarities between the target model's response and the target text. By approximating the gradient and using the importance score as the weight, the proposed method demonstrated solid results in exposing the safety concerns of several popular LVLMs. The proposed method is very practical since popular LVLM services often only provide the model's response.

**Strengths:**

- The threat model, in which the adversary can only get access to the model's response, is very realistic in the real-world setting since popular LVLMs are often offered as a service where only the model's response is available. Additionally, this paper studies the universal perturbation with targeted texts. Arguably, this type of attack is one of the most challenging ones.
- The proposed method is technically sound and comprehensive experiments demonstrate the empirical performance. All components of the proposed attack are well explained. Sufficient ablation study has been conducted in this paper.
- The safety concerns of LVLMs exposed in this paper could potentially have a large impact on the AI safety field.

**Weaknesses:**

- A large number of queries with similar contents might be easily detected by the defender. Experiments investigating potential defense could make the paper more comprehensive. An example of such detection is in [1].
- For comparison with CroPA in Table 4, the CroPA paper stated that they use OpenFlamingo rather than Flamingo. It would be good to specifically specify which model is being used for evaluation. If it is the open-source version, it would help to cite OpenFlamingo and clearly state that OpenFlamingo is used rather than Flamingo. If Table 4 uses Flamingo, it would not be appropriate to compare it with the results reported in CroPA in Table 4 since the open-source version has some slight differences from the original model.
- All the target texts are rather few words. It would be more comprehensive to include longer sentences. Will longer sentences increase the query budgets to achieve a similar performance as these short sentences?
- For targeted attacks, only reporting the semantic similarity scores makes it hard to fully understand the attack performance. Word/token-based evaluations, such as BLEU, are more appropriate.


It is not a weakness, but I believe the year of the publication for [45] in the reference is wrong. NeurIPS 2024 papers should be under review at the moment rather than published.

---

[1] Chen, Steven, Nicholas Carlini, and David Wagner. "Stateful detection of black-box adversarial attacks." In Proceedings of the 1st ACM Workshop on Security and Privacy on Artificial Intelligence, pp. 30-39. 2020.

**Questions:**

The reviewer's main concerns are these points and questions outlined in the weaknesses section. The reviewer suggests that the authors focus on these points in the rebuttal.

**Limitations:**

Limitations and potential negative impacts have been sufficiently discussed in the paper.

---

> ### Author Rebuttal · Authors · 2024-08-07
>
> **Q1: Experiments investigating potential defense could make the paper more comprehensive.**
>
> **A1:** Thanks for your suggestion. As shown in the following table, we evaluate the robustness of our adversarial patch with four popular defense methods. Specifically, PatchCleanser [a] is a state-of-the-art certifiable defense against adversarial patches. It uses double masking to certify the prediction. [c,d] are query-based defenses, which are specifically designed for detecting malicious queries by black-box attacks. [b] is reviewer's mentioned defense. It shows that our attack method is robust to them. We will add the experiments in the revision.
>
> |ASR against defense|defense [a]|defense [b]|defense [c]|defense [d]|
> |:----:|:----:|:----:|:----:|:----:|
> |Our attack|92%|86%|79%|75%|
>
> [a] PatchCleanser: Certifiably robust defense against adversarial patches for any image classifier. USENIX 2022
> [b] Stateful detection of black-box adversarial attacks. S&P 2020
> [c] Random noise defense against query-based black-box attacks. NeurIPS 2021
> [d] Blacklight: Scalable defense for neural networks against query-based black-box attacks. USENIX 2022
>
> **Q2: For comparison with CroPA in Table 4, the CroPA paper stated that they use OpenFlamingo rather than Flamingo. It would be good to specifically specify which model is being used for evaluation.**
>
> **A2:** We utilize the same open-source version for fair comparison in Table 4. We will revise the description and add the reference.
>
> **Q3: All the target texts are rather few words. It would be more comprehensive to include longer sentences. Will longer sentences increase the query budgets to achieve a similar performance as these short sentences?**
>
> **A3:** Since we evaluate the text semantic similarity to assign importance weights and assess the attack performance, our proposed attack is not sensitive to the length of the target response text. That means, longer sentences will not increase the query budgets and our pre-set query budget is sufficient to handle the cases. We conduct more experiments on longer sentences in the table below. We can find that our attack can also achieve great attack performance on long output targets. We will add more experiments and discussion in the revision.
>
> |Adversarial Target|ImageClassification|ImageCaption|VQA|Overall|
> |:----:|:----:|:----:|:----:|:----:|
> |I am sorry, there is no object you mentioned, so I cannot answer|0.837|0.814|0.865|0.839|
> |A beautiful bird with a black and white color in snow|0.819|0.842|0.848|0.836|
> |I want to destroy the whole world together|0.776|0.801|0.784|0.787|
> |A photo of a cat|0.862|0.841|0.854|0.852|
> |The view from the top of a hill overlooking the mountains|0.828|0.843|0.810|0.827|
>
> **Q4: For targeted attacks, only reporting the semantic similarity scores makes it hard to fully understand the attack performance.**
>
> **A4:** Thanks for your suggestion. Actually, in our paper, we follow the previous work [45] to utilize the semantic similarity for fair evaluation. This metric is more generalizable for evaluating different texts with the same semantics. For example, the output texts "unknown" and "I don't know" share similar semantics, they have relatively higher semantic similarity score but would achieve a low word-level metric. Of course, we can also utilize the *ExactMatch* and *Contain* metrics to conduct strict word-level evaluation on our attack to assess the attack success rates. Specifically, the *ExactMatch* metric determines whether the LVLM output exactly matches the predefined target text, whereas the *Contain* (similar to BLEU) metric checks whether the output contains the target text. This is especially useful when outputs exceed the predefined target length. We report the corresponding evaluations in the table below, where our method also achieves significant attack performance, demonstrating the effectiveness of our attack.
>
> |LVLM Model|Dataset|ImageClassification  (*ExactMatch*)|ImageClassification  (*Contain*)|ImageCaption (*ExactMatch*)|ImageCaption (*Contain*)|VQA (*ExactMatch*)|VQA (*Contain*)|
> |:----:|:----:|:----:|:----:|:----:|:----:|:----:|:----:|
> |LLaVA|MS-COCO|81.2%|83.7%|80.5%|82.1%|78.8%|83.7%|
> |LLaVA|DALLE-3|80.6%|84.3%|76.4%|82.9%|87.6%|90.1%|
> |LLaVA|VQAv2|81.0%|84.5%|77.2%|81.7%|85.4%|89.3%|
> |MiniGPT-4|MS-COCO|85.8%|86.1%|83.2%|85.6%|84.4%|88.5%|
> |MiniGPT-4|DALLE-3|79.4%|84.7%|84.3%|87.0%|85.9%|89.6%|
> |MiniGPT-4|VQAv2|84.8%|87.1%|85.9%|92.7%|87.3%|93.6%|
> |Flamingo|MS-COCO|80.5%|85.4%|77.9%|81.2%|79.4%|83.0%|
> |Flamingo|DALLE-3|82.4%|82.6%|84.5%|87.1%|80.3%|83.9%|
> |Flamingo|VQAv2|84.1%|85.3%|83.3%|86.7%|86.4%|89.2%|
> |BLIP-2|MS-COCO|81.6%|83.8%|76.4%|79.7%|82.5%|84.0%|
> |BLIP-2|DALLE-3|78.2%|81.9%|83.8%|84.3%|85.1%|85.4%|
> |BLIP-2|VQAv2|75.9%|79.6%|81.4%|84.5%|86.3%|88.2%|
>
> **Q5: It is not a weakness, but I believe the year of the publication for [45] in the reference is wrong.**
>
> **A5:** Thanks for your reminder. We will correct the details of reference [45] in the revision.

---

> > ### Comment · Area_Chair_jnMy · 2024-08-11
> >
> > Dear i8wm,
> >
> > What are your thoughts after reading the rebuttal and other reviews?
> >
> > Best,
> >
> > AC

---

> > > ### Comment · Reviewer_i8wm · 2024-08-12
> > >
> > > I appreciated the clarification made in the rebuttal. Overall, I think it is a technically solid paper with no major concerns with respect to evaluation and reproducibility. I would maintain my rating.

---

> > > > ### Author Response · Authors · 2024-08-13
> > > > **Thanks to the reviewer**
> > > >
> > > > We are very grateful to you for the high approval. Thank you once again for your valuable contributions.

---

### Official Review · Reviewer_K8hM · 2024-07-01

**Soundness:** 3
**Presentation:** 3
**Contribution:** 3
**Rating:** 5
**Confidence:** 4

**Summary:**

This research introduces a novel approach to creating universal adversarial attacks against Large Vision-Language Models (LVLMs).
It proposes the universal attacker against real-world LVLMs that operates with limited access to the model (only inputs and outputs).
The attack is designed to be task-agnostic, using a universal adversarial patch that can deceive LVLMs across various tasks and inputs.
The approach is model-agnostic, not requiring detailed knowledge of the LVLM's structure. Extensive experiments demonstrate the effectiveness of the attack against popular LVLMs like LLaVA, MiniGPT-4, Flamingo, and BLIP-2 across various tasks.

**Strengths:**

This paper addresses a significant gap in the field of adversarial attacks on LVLMs, offering a more practical and versatile approach compared to existing task-specific methods. The universal nature of the attack and its ability to function with limited model access make it particularly relevant for real-world applications and raise important questions about the security of LVLMs.

**Weaknesses:**

This paper presents an interesting approach to universal adversarial attacks on large vision-language models (LVLMs). However, there are several areas where the paper could be improved:

1. Comparison with non-universal attacks: The paper lacks a comparison between the proposed universal attack and non-universal attacks on specific tasks. Including such experiments would help readers better understand the advantages and limitations of this method. The current baselines ("Full attack" and "w/o importance") are insufficient, and more comparisons would strengthen the paper.

2. Query efficiency: The authors mention allowing 70,000 queries in total, but it's unclear how many images this applies to. This number seems excessive and impractical for real-world scenarios. A good attack should be query-efficient, and this aspect needs further explanation or justification.

3. Opposite directions claim: The statement about "some of them may have opposite directions" (line 205) lacks supporting evidence. This claim requires further explanation or empirical support.

4. Judge model acquisition: The paper should clarify how an attacker would obtain the judge model in practical scenarios and whether it needs to be related to the target LVLM.

5. Target text selection: Using "Unknown" as the target text may not truly represent a **targeted** attack. It would be valuable to include experiments with more specific target texts (e.g., "a photo of a cat") and report the success rates for such cases.

6. Patch visibility and physical attacks: Figure 4 suggests that the adversarial patch is quite noticeable to the human eye, which could be easily filtered in real-world applications. Additionally, the paper would benefit from including physical attack experiments, where the patch is printed and applied to real images to test LVLM performance.

7. Transferability to other models: The paper should explore whether this attack transfers to other image-capable language models like GPT-4 or Claude, as this would demonstrate the broader applicability of the method.

**Questions:**

see above

**Limitations:**

see above

---

> ### Author Rebuttal · Authors · 2024-08-07
>
> **Q1: Comparison with non-universal attacks.**
>
> **A1:** Since existing LVLM attackers are implemented in different settings with different models/datasets, we have already provided detailed comparisons in Table 3 and 4 of the paper. Note that, existing methods are non-universal attacks and require prior LVLM knowledge to generate different perturbations for different images. In contrast, our attack solely accesses to the LVLM input/output and can generate a single noised patch to fool all images/prompts, while achieving better attack performance. To provide a more comprehensive comparison, we also re-implement these attacks into our utilized datasets/models/metrics as shown in the table below:
>
> |LVLM Model|Attack Method|MS-COCO (Overall)|DALLE-3 (Overall)|VQAv2 (Overall)|
> |:----:|:----:|:----:|:----:|:----:|
> |LLaVA|MF-Attack|0.626|0.634|0.647|
> |LLaVA|CroPA|0.778|0.796|0.782|
> |LLaVA|Ours|**0.830**|**0.837**|**0.829**|
> |MiniGPT-4|MF-Attack|0.643|0.618|0.635|
> |MiniGPT-4|CroPA|0.803|0.780|0.819|
> |MiniGPT-4|Ours|**0.841**|**0.838**|**0.865**|
> |Flamingo|MF-Attack|0.671|0.654|0.662|
> |Flamingo|CroPA|0.796|0.825|0.814|
> |Flamingo|Ours|**0.835**|**0.844**|**0.850**|
> |BLIP-2|MF-Attack|0.639|0.658|0.670|
> |BLIP-2|CroPA|0.783|0.799|0.811|
> |BLIP-2|Ours|**0.814**|**0.824**|**0.836**|
>
> It shows that our attack is still more adversarial. We think the reason is that our universal patch explicitly learns the general adversarial patterns against LVLMs and effectively estimates gradients solely using positive directions.
>
> **Q2: Query efficiency.**
>
> **A2:** The overall 70k queries are utilized to generate the universal patch using 500 images (in Table 10 of the appendix B.5). This means 70k queries cover the processing of all 500 images, rather than each image requiring 70k individual queries. Moreover, as shown in Table 8 of the appendix B.3, our method takes only 4.9h to attack all images/prompts, which is significantly more efficient than previous methods that generate separate perturbations for each image/prompt, e.g., MF-Attack: 3h for only a single perturbation, CroPA: 2.6h for only a single perturbation.
>
> **Q3: Opposite directions claim.**
>
> **A3:** We have empirically supported this claim in Table 1 of the paper via the variant "w/o importance". In practice, not all slight additive noises point towards the optimal direction during the gradient estimation, as negative noise may contain decomposed components in the opposite direction. If we treat all noises equally, the "w/o importance" variant performs worse. However, by weakening the negative noise and strengthening the positive noise with positive/negative-aware importances, our full attack achieves the best performance. This demonstrates that not all noise contributes positively to the optimal gradient direction.
>
> **Q4: Judge model acquisition.**
>
> **A4:** Actually, the attackers can utilize any text embedding model as the judge model, like BERT, Sentence-BERT. As shown in Figure 5(a) of the paper, our attack is not sensitive to these text encoders. It does not need to be related to the target LVLM.
>
> **Q5: Target text selection.**
>
> **A5:** As shown in Table 2 and Figure 8 of the paper, our attack can achieve significant performance on specific target texts. We also conduct more experiments on such types of texts in the table below. We will add more experiments in the revision.
>
> |Adversarial Target|ImageClassification|ImageCaption|VQA|Overall|
> |:----:|:----:|:----:|:----:|:----:|
> |I am sorry, there is no object you mentioned, so I cannot answer|0.837|0.814|0.865|0.839|
> |A beautiful bird with a black and white color in snow|0.819|0.842|0.848|0.836|
> |I want to destroy the whole world together|0.776|0.801|0.784|0.787|
> |A photo of a cat|0.862|0.841|0.854|0.852|
> |The view from the top of a hill overlooking the mountains|0.828|0.843|0.810|0.827|
>
> **Q6: Patch visibility and physical attacks.**
>
> **A6:** Although the adversarial patch is noticeable to humans, it can effectively fool the LVLMs in the universal setting without using any model details compared to other global noise attacks (which are less obvious). Moreover, as shown in the following table, our adversarial patch is robust to existing defenses.
>
> |ASR against defense|defense [a]|defense [b]|defense [c]|defense [d]|
> |:----:|:----:|:----:|:----:|:----:|
> |Our attack|92%|86%|79%|75%|
>
> Besides, we try to implement two types of physical attack: A. we print and paste the patch on real images and scan it to query the LVLM; B. we print and paste the patch on a flat plane in realistic and collect new images. As shown in the table below, our method can still effectively attack these physical cases.
>
> |Adversarial Target|Case A (Overall)|Case B (Overall)|
> |:----:|:----:|:----:|
> |A photo of a cat|0.783|0.756|
>
> [a] PatchCleanser: Certifiably robust defense against adversarial patches for any image classifier. USENIX 2022
> [b] Stateful detection of black-box adversarial attacks. S&P 2020
> [c] Random noise defense against query-based black-box attacks. NeurIPS 2021
> [d] Blacklight: Scalable defense for neural networks against query-based black-box attacks. USENIX 2022
>
> **Q7: Transferability to other models.**
>
> **A7:** Firstly, we have tested the transferability among four LVLMs in Table 6 of the appendix B.1. Secondly, to evaluate the transfer-attack performance on GPT-4 and Claude, we also provide experiments in the following table. These experiments demonstrate the broader applicability of our attack.
>
> |Target Model|From LLaVA (Overall)|From MiniGPT-4 (Overall)|From Flamingo (Overall)|From BLIP-2 (Overall)|
> |:----:|:----:|:----:|:----:|:----:|
> |GPT-4|0.724|0.751|0.702|0.736|
> |Claude|0.743|0.744|0.718|0.725|

---

> > ### Comment · Area_Chair_jnMy · 2024-08-11
> >
> > Dear K8hM,
> >
> > What are your thoughts after reading the rebuttal and other reviews?
> >
> > Best,
> >
> > AC

---

> > ### Comment · Reviewer_K8hM · 2024-08-12
> > **after rebuttal**
> >
> > Thanks for the rebuttal! After reading the rebuttal, I would like to maintain my current score.

---

> > > ### Author Response · Authors · 2024-08-12
> > > **Thanks to the reviewer**
> > >
> > > Thank you for your valuable response. If you have any further concerns or questions, please feel free to contact us!

---

### Official Review · Reviewer_3jB6 · 2024-07-03

**Soundness:** 3
**Presentation:** 2
**Contribution:** 3
**Rating:** 5
**Confidence:** 4

**Summary:**

This paper presents a novel approach to create a universal adversarial attacker for LVLMs. The proposed method focuses on two main aspects: restricting access to only the LVLM inputs and outputs, and devising a task-agnostic adversarial patch that can deceive multiple multimodal downstream tasks. The approach involves initializing the adversarial patch through random sampling and optimizing it using a diverse set of LVLM task inputs. Extensive experiments demonstrate the effectiveness of the proposed method across various LVLMs and tasks.

**Strengths:**

1. The black-box attack design make the proposed method practical for real-world applications.
2. The universal adversarial patch can deceive multiple multimodal downstream tasks enhancing the proposed method's utility.
3.  Extensive experiments across different LVLMs and tasks validate the effectiveness of the proposed method.

**Weaknesses:**

1. The paper could benefit from clearer explanations of the technical details and processes involved in creating and optimizing the adversarial patch.

**Questions:**

I am curious about the output of the model for a completely corrupted input, such as an image with noise that cannot be recognized by a human. It would be interesting to include this as a baseline. The paper does not explicitly state this nuance, but it seems this attack aims to mislead the model with an image that is clearly recognizable to humans. Otherwise, one could always send the model an image with pure noise, which would not be a useful attack.

---

> ### Author Rebuttal · Authors · 2024-08-07
>
> **Q1: The paper could benefit from clearer explanations of the technical details and processes.**
>
> **A1:** Thanks for your suggestion. We will add more corresponding explanations in the revision:
>
> (1) For adversarial patch initialization: to achieve successful targeted attacks, there is no prior knowledge of where the patch shall be located and what it looks like. Therefore, for each targeted text, we randomly sample a few patch locations and patterns, and choose a combination of the two with the highest attack performance. In particular, to determine whether a location is good, we randomly sample a set of patterns at that location and examine the average attack performance for this set. Since the quality of a patch location also relies on sampled patterns, we also draw patterns from a uniform distribution to find the optimal patch pattern.
>
> (2) For adversarial patch optimization: Since there is no model gradients or details, we need to estimate the gradient direction for further perturbing the patch pattern. The general Monte Carlo estimations apply a set of random noises on the patch and see whether these noises can change the prediction. These perturbation directions are averaged as the final direction for mutating the patch. However, such a design is not efficient as not all noises point towards optimal direction. To this end, we propose an importance-aware gradient approximation method to adaptively adjust the weights for different samples, based on how these sampled noises can lead to the attacker-chosen text. Specifically, given a set of slight additive noise in each step, we first add them to the patch to assess their positive/negative degrees via our developed LVLM-aware indicator function. Based on this, we then assign corresponding weights to positive/negative noises to strengthen the positive gradient direction while weakening the negative gradient direction. In this manner, we can efficiently and effectively optimize the adversarial patch with optimal estimated gradients.
>
> **Q2: I am curious about the output of the model for a completely corrupted input, such as an image with noise that cannot be recognized by a human. It would be interesting to include this as a baseline.**
>
> **A2:** We have conducted detailed comparisons between our adversarial patch and completely corrupted inputs in Table 3,4 of our paper. In these two tables, although previous LVLM attackers (MF-Attack, CroPA) aim to globally perturb the whole image to improve the imperceptibility, they rely on the LVLM details and gradients for perturbation updation. Instead, our adversarial patch can be optimized in a more challenging but practical setting by solely querying the model, while achieving better attack performance than these LVLM attackers (Ours vs MF-Attack: 0.727 vs 0.646; Ours vs CroPA: 0.69 vs 0.65), demonstrating our effectiveness. Besides, we also try to re-implement these completely corrupted attacks into our utilized datasets/models/metrics as baselines for comparison in the following table. It shows that our attack is still more adversarial as we explicitly and effectively estimate optimal gradient direction for optimizaiton.
>
> |LVLM Model|Attack Method|MS-COCO (Overall)|DALLE-3 (Overall)|VQAv2 (Overall)|
> |:----:|:----:|:----:|:----:|:----:|
> |LLaVA|MF-Attack|0.626|0.634|0.647|
> |LLaVA|CroPA|0.778|0.796|0.782|
> |LLaVA|Ours|**0.830**|**0.837**|**0.829**|
> |MiniGPT-4|MF-Attack|0.643|0.618|0.635|
> |MiniGPT-4|CroPA|0.803|0.780|0.819|
> |MiniGPT-4|Ours|**0.841**|**0.838**|**0.865**|
> |Flamingo|MF-Attack|0.671|0.654|0.662|
> |Flamingo|CroPA|0.796|0.825|0.814|
> |Flamingo|Ours|**0.835**|**0.844**|**0.850**|
> |BLIP-2|MF-Attack|0.639|0.658|0.670|
> |BLIP-2|CroPA|0.783|0.799|0.811|
> |BLIP-2|Ours|**0.814**|**0.824**|**0.836**|
>
> Secondly, previous completely corrupted inputs are diverse and sensitive to their global noises. These noises are specific to different images or prompts and can hardly to be effective in attacking other images/prompts. Instead, as shown in Figure 3 of the paper, our adversarial patch design is more flexible and can easily achieve a more challenging universal attack.
>
> **Q3: The paper does not explicitly state this nuance, but it seems this attack aims to mislead the model with an image that is clearly recognizable to humans.**
>
> **A3:** Thanks for your concern. We want to clarify that:
>
> (1) Our main goal is to tackle two critical issues in existing LVLM attackers (not to mislead LVLMs with recognizable perturbations): (i) restricting access to only the LVLM inputs and outputs. (ii) devising a universal adversarial perturbation.
>
> (2) To achieve the above goal, traditional attacks using completely corrupted inputs are ineffective as they require backpropagated gradients for optimization and their perturbations vary across different images/prompts. Therefore we choose to use adversarial patch to hanlde the above two challenging issues, which has been proven to be more effective than previous global noises through our carefully designed patch initialization and optimization processes.
>
> (3) We can further improve the imperceptibility of our attack by using smaller patch, which also achieves competitive performance as in Table 5 of the paper. Besides, our patch pattern is well-optimized via our designed strategy to achieve the challenging targeted attack, which is quite different from ineffective random pure noise.
>
> (4) Moreover, as in the following table, our adversarial patch is robust to existing defenses.
>
> |ASR against defense|defense [a]|defense [b]|defense [c]|defense [d]|
> |:----:|:----:|:----:|:----:|:----:|
> |Our attack|92%|86%|79%|75%|
>
> [a] PatchCleanser: Certifiably robust defense against adversarial patches for any image classifier. USENIX 2022
> [b] Stateful detection of black-box adversarial attacks. S&P 2020
> [c] Random noise defense against query-based black-box attacks. NeurIPS 2021
> [d] Blacklight: Scalable defense for neural networks against query-based black-box attacks. USENIX 2022

---

> > ### Comment · Area_Chair_jnMy · 2024-08-11
> >
> > Dear 3jB6,
> >
> > What are your thoughts after reading the rebuttal and other reviews?
> >
> > Best,
> >
> > AC

---

### Official Review · Reviewer_8diJ · 2024-07-11

**Soundness:** 3
**Presentation:** 3
**Contribution:** 3
**Rating:** 5
**Confidence:** 4

**Summary:**

The paper proposes a universal adversarial attack method targeting large vision-language models (LVLMs) by designing a universal adversarial patch. This method restricts access to only the model’s inputs and outputs, creating a task-agnostic adversarial patch that can deceive various LVLM-driven tasks without prior model knowledge. Extensive experiments validate the effectiveness of the proposed attack across multiple models and tasks.

**Strengths:**

1. This paper investigates the vulnerability of real-world LVLMs in a practical but challenging setting, where 83 the attackers can only access the input and output of the LVLM. It is more practical.
2. This paper devises a universal adversarial patch that can be pasted and then fool any inputs for any LVLM downstream task, it is significant to evaluate the security of LVLMs, especially in the security-critical scenarios.

**Weaknesses:**

1. In this attack setting, constraints don't seem as important because there is no need to ensure stealthiness, but patch size is quite crucial. In extreme cases, if the patch size is set to be as large as the image, the attack loses its meaning. So, can the attack achieve the same effect with a very small patch size without adding constraints to the perturbation?
2. Why do different target labels affect the choice of patch position? Can the patch be fixed in a specific position to achieve the same attack effect?
3. Compared to adversarial example attacks, this type of attack is certainly more powerful. In practical scenarios, this attack seems more meaningful for jailbreaking large models, such as inducing the model to follow harmful instructions or provide harmful information. The responses tested in the paper are relatively short, such as a single word. Does the length of the target response text affect the effectiveness of this attack?
4. In real-world scenarios, since this patch attack is quite obvious, a user could simply remove the patch's effect by cropping the image. So, what is the practical significance of this type of attack?

**Questions:**

Refer to the Weakness section.

**Limitations:**

No.

---

> ### Author Rebuttal · Authors · 2024-08-07
>
> **Q1: Can the attack achieve the same effect with a very small patch size without adding constraints to the perturbation?**
>
> **A1:** Without adding perturbation constraints, a smaller patch can achieve a similar attack performance. We implement this perturbation constraint to make a fair comparison with existing LVLM attack methods, as shown in Table 3 and 4 of the paper. Although we have conducted ablation studies on different patch sizes in Table 5 of the paper, these experiments are carried out with the pre-set perturbation constraint. We provide ablations without adding perturbation constraints in the following table:
>
> |Patch Size|ImageClassification|ImageCaption|VQA|Overall|
> |:----:|:----:|:----:|:----:|:----:|
> |S_p=16 (w/o. constraint)|0.776|0.753|0.825|0.785|
> |S_p=32 (w/. constraint)|0.734|0.720|0.778|0.744|
> |S_p=32 (w/o. constraint)|0.805|0.787|0.842|0.811|
> |S_p=48 (w/. constraint)|0.793|0.775|0.842|0.803|
> |S_p=48 (w/o. constraint)|0.819|0.823|0.860|0.834|
> |S_p=64 (w/. constraint)|0.824|0.806|0.879|0.837|
> |S_p=64 (w/o. constraint)|0.858|0.842|0.901|0.867|
>
> It shows that, without constraint, patch size 48x48 (w/o. constraint) can achieve a similar overall adversarial effect as our main attack setting of patch size 64x64 (w/. constraint) in the paper (0.834 vs 0.837). Moreover, patch size 32x32 without the constraint also outperforms its constrained variant, and even a much smaller patch size 16x16 without the constraint also achieves a very competitive attack performance. This highlights the significant potential of our proposed attack.
>
> **Q2: Why do different target labels affect the choice of patch position? Can the patch be fixed in a specific position to achieve the same attack effect?**
>
> **A2:** (1) We attribute this to the fact that different local regions of the same image contribute varying levels of ambiguity to different target labels, which affects the LVLM model’s ability to output specific semantic text. For instance, in Figure 4 of the paper, "I cannot answer" and "I am sorry" have similar semantics and therefore share close patch positions, while the distinct semantic "I hate people" corresponds to a different position.
>
> (2) Moreover, we investigate the performances of a fixed patch position (on the upper left corner of images) in the following table.
>
> |Adversarial Target|Overall (w/o. fixed patch)|Overall (w/. fixed patch)|
> |:----:|:----:|:----:|
> |Unknown|0.837|0.796|
> |I cannot answer|0.844|0.815|
> |I am sorry|0.842|0.807|
>
> This shows that a fixed position can degrade attack performance, indicating that the initial patch position is crucial for targeting specific texts. However, the fixed patch still exhibits a competitive adversarial attack effect, demonstrating the generalizability of our approach.
>
> **Q3: Does the length of the target response text affect the effectiveness of this attack?**
>
> **A3:** Since we evaluate the semantic similarity to assess the attack performance in the judge model and indicator function, our attack is not sensitive to the length of the target response text. As shown in Table 2 and Figure 8 of the paper, our attack can also achieve significant attack performance on long target text. We also conduct more experiments on longer target text in the table below:
>
> |Adversarial Target|ImageClassification|ImageCaption|VQA|Overall|
> |:----:|:----:|:----:|:----:|:----:|
> |I am sorry, there is no object you mentioned, so I cannot answer|0.837|0.814|0.865|0.839|
> |A beautiful bird with a black and white color in snow|0.819|0.842|0.848|0.836|
> |I want to destroy the whole world together|0.776|0.801|0.784|0.787|
> |A photo of a cat|0.862|0.841|0.854|0.852|
> |The view from the top of a hill overlooking the mountains|0.828|0.843|0.810|0.827|
>
> It shows that the length of target texts does not affect the attack effectiveness, demonstrating the scalability of our attack.
>
> **Q4: What is the practical significance of this type of attack?**
>
> **A4:** The practical significance of our attack lies in four-fold:
>
> (1) In practical cases, our patches have great potential to be pasted on the realistic images/pictures to achieve attacks [e,f]. We implement two types of physical attack: A. we print and paste the patch on real images and scan it to query the LVLM; B. we print and paste the patch on a flat plane in realistic and collect new images. As in the table below, our method can effectively attack these physical cases.
>
> |Adversarial Target|Case A (Overall)|Case B (Overall)|
> |:----:|:----:|:----:|
> |A photo of a cat|0.783|0.756|
>
> (2) To reduce our adversarial effect, directly cropping it may destroy the image structure and lose some contents. Moreover, as shown in the following table, our attack is robust to existing detection-based defenses, indicating that it is hard for automatic systems to prevent our attack.
>
> |ASR against defense|defense [a]|defense [b]|defense [c]|defense [d]|
> |:----:|:----:|:----:|:----:|:----:|
> |Our attack|92%|86%|79%|75%|
>
> (3) Our model is practical in real-world LVLM applications as we solely access to the models' input/output. However, existing LVLM attacks severely rely on the prior LVLM model knowledge.
>
> (4) Our universal patch design also demonstrates our scalability. Compared to previous adversarial perturbation attacks, our attack is efficient and effective to fool different images/prompts in a single process.
>
> [a] PatchCleanser: Certifiably robust defense against adversarial patches for any image classifier. USENIX 2022
> [b] Stateful detection of black-box adversarial attacks. S&P 2020
> [c] Random noise defense against query-based black-box attacks. NeurIPS 2021
> [d] Blacklight: Scalable defense for neural networks against query-based black-box attacks. USENIX 2022
> [e] Naturalistic Physical Adversarial Patch for Object Detectors. ICCV 2021
> [f] DAP: A Dynamic Adversarial Patch for Evading Person Detectors. CVPR 2024

---

> > ### Comment · Area_Chair_jnMy · 2024-08-11
> >
> > Dear 8diJ,
> >
> > What are your thoughts after reading the rebuttal and other reviews?
> >
> > Best,
> >
> > AC

---

> > ### Comment · Reviewer_8diJ · 2024-08-12
> >
> > My concerns are properly addressed.
> >
> > Thanks for the rebuttal!

---

> > > ### Author Response · Authors · 2024-08-12
> > > **Thanks to the reviewer**
> > >
> > > We would like to thank the reviewer for responding to our rebuttal. It is great to know that your concerns have been addressed.

---

### Author Rebuttal · Authors · 2024-08-07

Dear reviewers,

We much appreciate for your acknowledgment of our work and helpful, insightful comments. Following the reviewers' suggestions, we have carefully revised the paper and conducted a series of new experiments to address the reviewers' concerns. In the following, under each reviewer's comment, we address the concerns of the reviewers point by point.

---

### Decision · Program_Chairs · 2024-09-25

**Decision:**

Accept (poster)

**Comment:**

The paper proposes a universal adversarial attack method for Large Vision-Language Models (LVLMs). Specifically, they generate a task-agnostic adversarial patch attacking multiple black-box LVLMs. Overall, the reviewers acknowledge the value of the paper, especially after the rebuttal.